# Increased genetic contribution to wellbeing during the COVID-19 pandemic

C. A. Robert Warmerdam[1⊙], Henry H. Wiersma[1⊙], Pauline Lanting[1⊙], Alireza Ani[2], Lifelines Corona Research Initiative[¶], Lifelines Cohort Study[¶], Marjolein X. L. Dijkema[1], Harold Snieder[2], Judith M. Vonk[2], H. Marike Boezen[2†], Patrick Deelen[1,3], Lude H. Franke[1,3]*

1 Department of Genetics, University Medical Center Groningen, University of Groningen, Groningen, The Netherlands, 2 Department of Epidemiology, University Medical Center Groningen, University of Groningen, Groningen, The Netherlands, 3 Oncode Institute, Utrecht, The Netherlands

⊙ These authors contributed equally to this work.
† Deceased.
¶ Membership of the Lifelines Corona Research Initiative and Lifelines Cohort Study are provided in the Acknowledgments.
* l.h.franke@umcg.nl

## Abstract

Physical and mental health are determined by an interplay between nature, for example genetics, and nurture, which encompasses experiences and exposures that can be short or long-lasting. The COVID-19 pandemic represents a unique situation in which whole communities were suddenly and simultaneously exposed to both the virus and the societal changes required to combat the virus. We studied 27,537 population-based biobank participants for whom we have genetic data and extensive longitudinal data collected via 19 questionnaires over 10 months, starting in March 2020. This allowed us to explore the interaction between genetics and the impact of the COVID-19 pandemic on individuals' wellbeing over time. We observe that genetics affected many aspects of wellbeing, but also that its impact on several phenotypes changed over time. Over the course of the pandemic, we observed that the genetic predisposition to life satisfaction had an increasing influence on perceived quality of life. We also estimated heritability and the proportion of variance explained by shared environment using variance components methods based on pedigree information and household composition. The results suggest that people's genetic constitution manifested more prominently over time, potentially due to social isolation driven by strict COVID-19 containment measures. Overall, our findings demonstrate that the relative contribution of genetic variation to complex phenotypes is dynamic rather than static.

## Author summary

All over the world we have experienced the influence of the COVID-19 pandemic on our wellbeing. However, the impact may not have been the same for everyone. We know that physical and mental health are affected partly by nature, for example genetics, and partly by environmental factors, for example the COVID-19 pandemic. Here, we explored the

**Data Availability Statement:** Results were frequently shared with participants and the general public through interactive infographics on the

Corona Barometer website (https://coronabarometer.nl/). The individual-level data that support the findings in this publication were obtained from the Lifelines biobank under project application number ov20_0554. Due to privacy reasons the individual-level data can't be made publicly available but can be made available upon reasonable request. This request should be directed to the Lifelines Research Office through email (research@lifelines.nl) or by using the application form on their website (https://www.lifelines.nl/researcher/how-to-apply/apply-here). All code that is central to this paper is made available via GitHub (https://github.com/molgenis/covid19_prs_time). Other analyses of genotype data and publicly available reference data is performed using standard bioinformatics practices, for which the code is made available upon request.

**Funding:** LHF is supported by grants from the Dutch Research Council (ZonMW-VIDI 917.14.374 and ZonMW-VICI 09150182010019 to LHF) and by an ERC Starting Grant, grant agreement 637640 (ImmRisk) and through a Senior Investigator Grant from the Oncode Institute. PD is supported by a grant from the Dutch Research Council (ZonMW-VENI 9150161910057 to PD). The Lifelines Biobank initiative has been made possible by funding from the Dutch Ministry of Health, Welfare and Sport, the Dutch Ministry of Economic Affairs, the University Medical Center Groningen (UMCG the Netherlands), the University of Groningen, the Northern Provinces of the Netherlands, FES (Fonds Economische Structuurversterking), SNN (Samenwerkingsverband Noord Nederland) and REP (Ruimtelijk Economisch Programma). The generation and management of GWAS genotype data for the Lifelines Cohort Study is supported by the UMCG Genetics Lifelines Initiative (UGLI) and by a Spinoza Grant from NWO, awarded to Cisca Wijmenga. The funders had no role in study design, data collection and analysis, decision to publish, or preparation of the manuscript.

**Competing interests:** I have read the journal's policy and the authors of this manuscript have the following competing interests: Author H. Marike Boezen was unable to confirm their authorship contributions. On their behalf, the corresponding author has reported their contributions to the best of their knowledge. The authors declare no further competing interests.

interaction between genetics and the impact of the COVID-19 pandemic on individuals' wellbeing over time. We observed that genetics not only influenced many aspects of wellbeing, but also that this impact changed over time during the pandemic. Our results suggest that the relative contribution of an individuals' genetics increased over time. Overall, our findings demonstrate that the relative contribution of genetic variation to complex phenotypes, such as wellbeing, is dynamic rather than static.

## Introduction

Early 2020, life all over the world was dramatically impacted by the COVID-19 pandemic, which represents a unique situation in which whole communities were suddenly and simultaneously exposed to both the virus and the societal changes required to combat the virus. In response to the pandemic the Lifelines COVID-19 cohort was initiated within the Lifelines prospective follow-up biobank. Online questionnaires were sent out to participants starting March 30th 2020, to investigate the (genetic) risk factors for COVID-19 and its health and societal impacts, including a wide variety of physical and mental health and lifestyle behaviour phenotypes [1,2].

Since the start of the pandemic, the genetics of COVID-19 and the impact of the COVID-19 pandemic on mental health over time have been extensively investigated separately [3,4]. Here, we aimed to explore the interaction between genetics and the impact of the COVID-19 pandemic on individuals' wellbeing over time. Physical and mental health are known to be determined by an interplay between nature, for example genetics, and nurture, which encompasses experiences and exposures that can be short or long-lasting. Health conditions of individuals are also often the product of interplay between genetics and the environment. Depressive episodes, for example, are partly the result of an interaction between stressful life-events and a genetic predisposition to depression [5–8]. We hypothesize that the COVID-19 pandemic is a potentially traumatic life-event, having previously been associated to highly significant levels of psychological distress [9–12], and therefore expect to observe a dynamic relative contribution of genetics on wellbeing during the pandemic.

## Results and discussion

To explore the role of genetics in individual's experiences of the COVID-19 pandemic, we studied the impact of genetic variation on physical and mental health and lifestyle behaviours in the Lifelines biobank, a prospective follow-up cohort study of 167,000 participants living in the three northern provinces of the Netherlands. The repeated questionnaires allowed us to longitudinally track physical and mental health and lifestyle behaviours, and we report here on results for the 19 questionnaires sent over the first 10 months of the project (Fig 1, and S1 and S2 Tables). In total, we had genotype data for 27,537 participants of whom 17,831 had completed at least one questionnaire during the first half of the study period as well as at least one questionnaire during the second half of our study (S1 Fig). The 17,831 participants with longitudinal data completed on average 13 questionnaires. We used the genetic data to calculate 17 polygenic scores (PGSs) for each participant based on summary statistics of genome-wide association studies (GWASs) for BMI [13], COVID-19 susceptibility and severity [3], educational attainment [14], life satisfaction [15], personality traits [16,17], behavioural traits [18–20] and psychiatric diseases [21–24] (S3 and S4 Tables and S2 Fig). We also estimated heritability and the proportion of variance explained by shared environment using variance components methods based on Lifelines pedigree and household data.

**a)** # lifelines

Prospective follow-up cohort study in 167,000 participants, living in the three northern provinces of the Netherlands (Friesland, Groningen, Drenthe)

**Lifelines Corona** Research

Lifelines Corona Research questionnaires focuses on:

Physical health   Mental health   Infection characteristics

The Netherlands

Groningen

Friesland

Drenthe

**b)** 50,802 participants genotyped on Global Screening Array (GSA) or HumanCytoSNP-12

76,257 participants returned at least one of 19 corona questionnaires

139,713 participants approached
19 questionnaires (March 2020 - January 2021)

30 Mar 2020 / 6 Apr 2020 / 13 Apr 2020 / 20 Apr 2020 / 27 Apr 2020 / 4 May 2020 / 18 May 2020 / 1 Jun 2020 / 15 Jun 2020 / 6 Jul 2020 / 13 Jul 2020 / 10 Aug 2020 / 07 Sep 2020 / 12 Oct 2020 / 2 Nov 2020 / 17 Nov 2020 / 30 Nov 2020 / 14 Dec 2020 / 11 Jan 2021

Calculate polygenic scores

**c)** 17 traits with GWAS summary statistics for calculating polygenic scores

| General phenotypes |
| --- |
| BMI |
| Educational attainment |
| Life satisfaction |
| COVID-19 severity |
| COVID-19 susceptibility |
| **Personality traits** |
| Anxiety/tension |
| Worry/vulnerability |
| Neuroticism |
| **Behavioural traits** |
| Physical activity (2) |
| General risk tolerance |
| Tobacco use |
| Alcohol use |
| **Psychiatric traits** |
| Schizophrenia |
| Depression |
| Autism |
| Obsessive compulsive disorder |

**d)** Study relation between polygenic scores and 288 outcome items

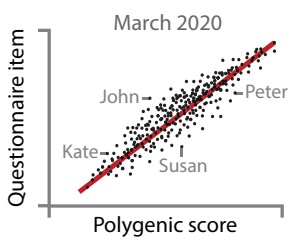

March 2020

John → ← Peter

Kate → ↑ Susan

Questionnaire item / Polygenic score

**e)** Study temporal aspect of genetic effect on 17 outcome items

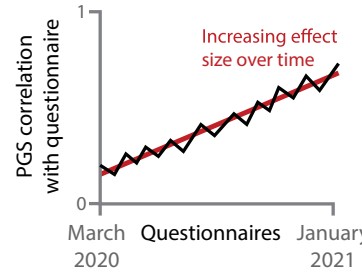

Increasing effect size over time

PGS correlation with questionnaire

March 2020   Questionnaires   January 2021

Genotyped Lifelines corona questionnaire respondents

| | Max. no. of participants |
| --- | --- |
| Global screening array | 19,703 |
| HumanCytoSNP-12 | 7,834 |
| Total | 27,537 |

Participants with response before and after August 31st

| | Max. no. of participants |
| --- | --- |
| Global screening array | 12,541 |
| HumanCytoSNP-12 | 5,290 |
| Total | 17,831 |

**Fig 1. Overview of the Lifelines COVID-19 cohort.** a) Our study is based on the Lifelines prospective follow-up cohort study and the accompanying Lifelines Corona Research project in which we sent out repeated questionnaires. b) We used 27,537 participants from European ancestry for whom we have genotype data available. c) Using the results of genome-wide association studies, we calculated polygenic scores (PGSs) for 17 traits. d) The relation between 17 PGSs and 288 outcomes was assessed using linear, logistic and ordered logistic regression models. e) Repeated questionnaire items allow us to study the temporal variation in the contribution of genetics to each of 46 PGS-outcome pairs using a longitudinal mixed-effects model and heritability analysis. The base layer of the map in panel a was retrieved from: geodata.nationaalgeoregister.nl/cbsgebiedsindelingen/wfs?request=GetFeature&service=WFS&version=2.0.0&typeName=cbs_provincie_2021_gegeneraliseerd&outputFormat=json.

## Baseline association analyses

Within the 27,537 samples with genotype data available we performed a baseline association analyses between the 17 PGSs and the responses to 288 questionnaire and questionnaire derived outcome items (S5 Table). We observed 302 Bonferroni-corrected significant associations for 143 unique outcomes (Figs 2 and S3, and S6 and S7 Tables) (Bonferroni-corrected $\alpha \leq 0.05$). This indicates a genetic influence on the features measured by the corresponding questionnaire items, which covered aspects of mental health, attitudes towards pandemic public health measures, COVID-19 exposures and cases, and physical health. For instance, we observe that high PGSs for Neuroticism, Schizophrenia and Depression are positively associated with outcome items about personality traits, health complaints, fatigue, and exhaustion (i.e. Neuroticism-PGS and "felt nervous", p-value = $1.88\times10^{-23}$; Neuroticism-PGS and "felt tired quickly", p-value = $2.65\times10^{-16}$), whereas we saw the opposite direction of association for the Life satisfaction-PGS where a high PGS is associated with lower "chest pain" scores (p-value = $4.48\times10^{-10}$). The PGSs for Life satisfaction and Neuroticism are also, respectively, negatively and positively associated to other health complaints and wellbeing items (i.e., Neuroticism-PGS and "excessive worrying", p-value = $6.75\times10^{-13}$; Life satisfaction-PGS and "quality of life"; p-value = $8.13\times10^{-14}$). The observation that people with a high genetic burden for psychological traits score lower on wellbeing related questions is in concordance with previous literature [25–27].

There are also PGSs that are significantly associated to the outcomes of several COVID-19–related questions. For instance, we observe many significant associations between the PGS for Educational attainment and variables pertaining to COVID-19 developments, including opinions about these developments (e.g., "having trust in the government's response to the COVID-19 pandemic"; p-value = $1.05\times10^{-9}$) and public health measures (e.g., "not shaking hands"; p-value = $6.39\times10^{-7}$). Furthermore, we observed positive associations between the PGS for Worry/Vulnerability and "worry about infecting someone else" (p-value = $2.41\times10^{-6}$), and between the PGS for Alcohol consumption and "whether or not individuals avoid facilities like bars and restaurants as a COVID-19 precaution" (p-value = $9.65\times10^{-8}$). In November 2020, individuals with a genetic predisposition for risky behaviour indicated they were more often planning to travel outside the country to go skiing/snowboarding (p-value = $7.73\times10^{-07}$), although the opportunity ultimately did not arise due to COVID-19 related restrictions. This shows that existing PGSs can also inform on responses that are specific to the current pandemic. Indicating that the behaviour of individuals during the pandemic can be influenced by known behavioural genetic factors that are captured by the PGSs from existing GWASs.

## Longitudinal analyses

Most GWASs are conducted at a single point in time and identify a static relationship between genetics and a trait. Due to the synchronized and prolonged exposure of the COVID-19 pandemic we were able to determine if this genotype-phenotype relationship can change over time. To do so we performed longitudinal analyses within 17,831 samples using mixed-effect

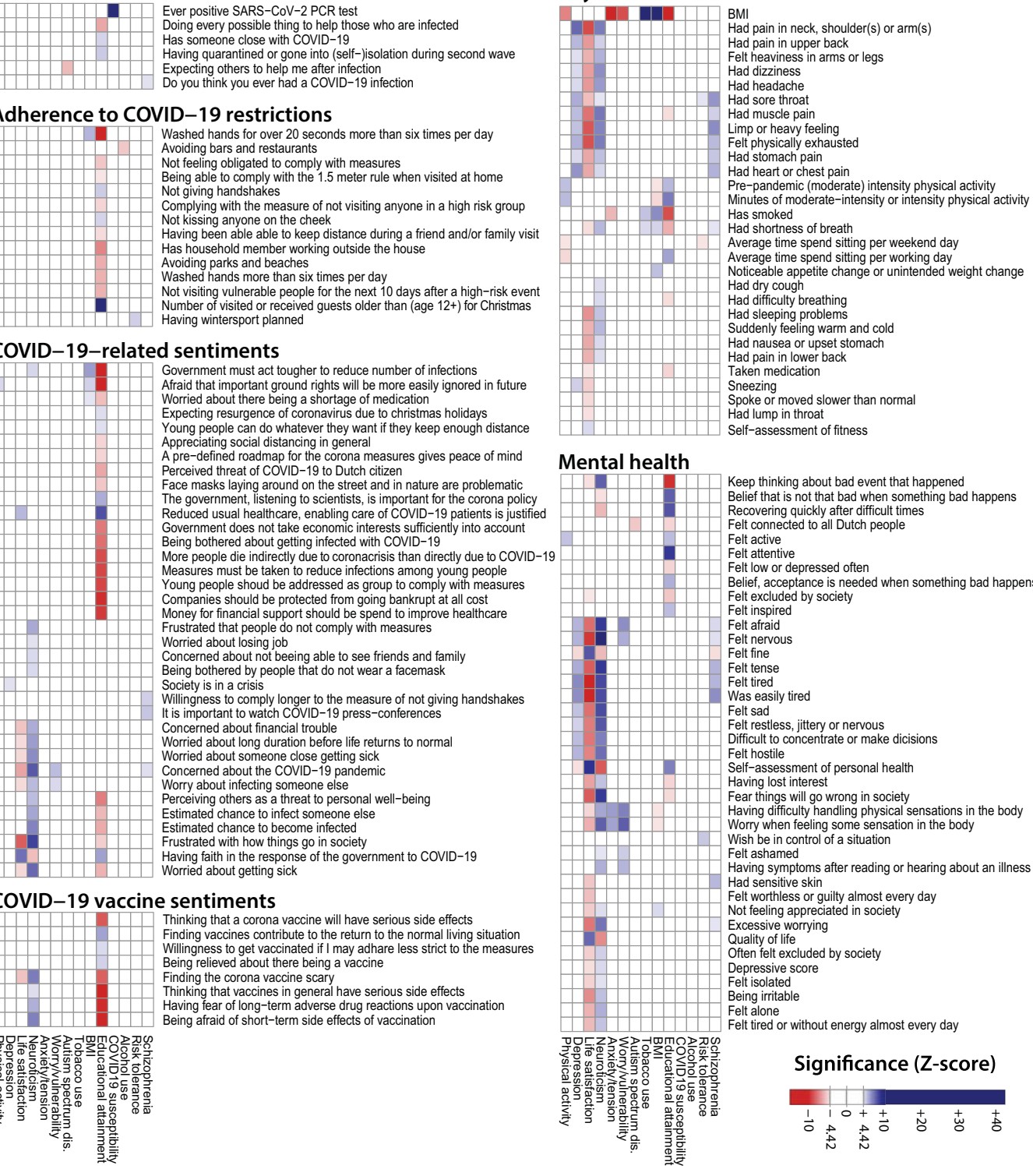

**Fig 2. Significant associations of PGSs and question answers at baseline.** A Z-score heatmap highlighting significant associations between questionnaire outcome items and polygenic scores (PGSs). In total, we analysed 288 questionnaire items and questionnaire-derived items, of which 143 have a significant association with at least one PGS. We observe significant associations of the PGSs for Depression, Life satisfaction, Neuroticism and Schizophrenia with wellbeing. Additionally, COVID-19 sentiments are associated to PGS traits including Educational attainment, Neuroticism and Worry/Vulnerability. Associations were estimated between PGS and outcome items using a multivariate model (linear, logistic, or ordered logistic regression) with age, sex, household size, having children and having chronic disease used as covariates. The 'Ever positive SARS-CoV-2' item was determined from answers to all

questionnaires. For all other items we used the first time the question was asked. We used two PGSs related to physical activity (Accelerometer-based physical activity & Moderate to vigorous physical activity). For clarity we used the maximum Z-score of these two PGSs for the physical activity column in this figure.

models for 46 PGS–question pairs that had a significant baseline association and that have been asked at multiple timepoints (S8 Table). Fourteen of these showed a time dependent effect at nominal significance (p-value ≤ 0.05). At a false-discovery rate (FDR) of 0.05, 11 PGS–question pairs showed a significant time dependent effect (including the PGSs for Life satisfaction, Neuroticism, Depression, Schizophrenia, and COVID-19 susceptibility, S4 Fig and S9 Table and S1 Note), of which two were Bonferroni significant: genetic predisposition of COVID-19 susceptibility with a positive SARS-CoV-2 PCR test and the PGS for Life satisfaction with "felt tired".

**An increased effect of genetics on wellbeing over time.** We found that the PGSs for Life satisfaction, Neuroticism and Depression affected five correlated outcomes related to wellbeing (S4 Fig and S9 Table). We observed time-dependent effects for "perceived quality of life", "feeling good", "was easily tired", "feeling tired" and "feeling physically exhausted". Since the PGSs of these traits, as well as the answers to the questions, are either positively or negatively correlated to each other (S1 and S5 Figs), we assume that this is a single effect. In the interest of clarity, we focus our discussion below on the effect of the Life satisfaction-PGS on "perceived quality of life".

The mean perceived quality of life varied over time, with a peak during the summer of 2020 that was likely due to summer holidays and warm weather, but also because COVID-19-related restrictions and infections were at a minimum at that time (Figs 3, S6 and S7 and S10 Table). As expected, we observed that the PGS for Life satisfaction is positively associated with "perceived quality of life" (p-value = $8.13 \times 10^{-14}$).

What is intriguing is that the effect of the PGS for Life satisfaction on the perceived quality of life increased across the pandemic (p-value = $3.1 \times 10^{-3}$) (Fig 3). At the end of the summer, the mean perceived quality of life started to decline, but participants with a higher Life satisfaction-PGS appear to be more resilient and reported smaller decreases in quality of life, while participants with a lower PGS reported a stronger decrease in their reported quality of life. This shows that genetic predisposition has increased in importance over the course of the pandemic.

There are several possible social and psychological explanations for the increasing influence of genetics on the reported quality of life over time. One possibility is that, due to lockdown restrictions, people had fewer social contacts, which are known to affect wellbeing [28]. This social isolation could explain a diminished environmental influence on wellbeing relative to the genetic component. Alternatively, traumatic events can trigger depressive episodes in people with a genetic predisposition for depression and people who have a strong neurotic personality respond more strongly to stressors [29,30]. Since the Life satisfaction-PGS is negatively correlated with the Depression-PGS (Pearson r: -0.64 p-value: $\leq 2.22 \times^{-308}$) and Neuroticism-PGS (Pearson r: -0.71 p-value: $\leq 2.22 \times^{-308}$), this could potentially explain people's reduced resilience in the presence of the prolonged stress caused by the pandemic. This is also consistent with our finding that people with a high PGS for Depression or Neuroticism report being more tired, an effect that also increased in strength as the pandemic progressed. Either of these factors–social isolation or traumatic impact–could explain why people with a lower PGS for Life satisfaction had more difficulty dealing with the COVID-19 pandemic.

An alternate possibility is that the arrival of COVID-19 and the accompanying lockdown measures had a very strong impact on the quality of life of the society, such that the contribution of genetics on perceived quality of life was suddenly much smaller. Over time, with a

**a) Mean quality of life**

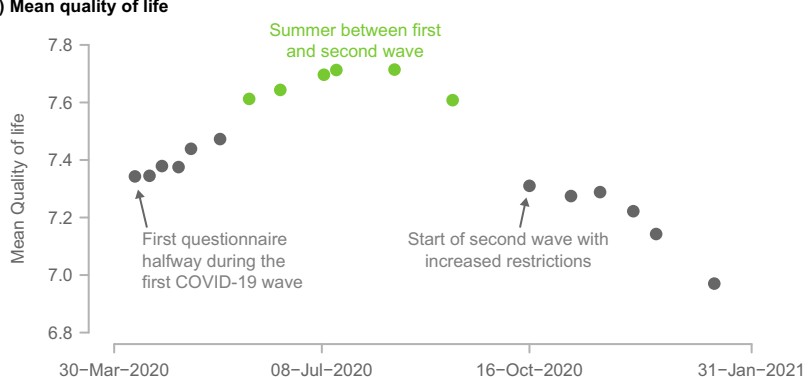

**b) Fitted model in 12,522 global screening array samples**

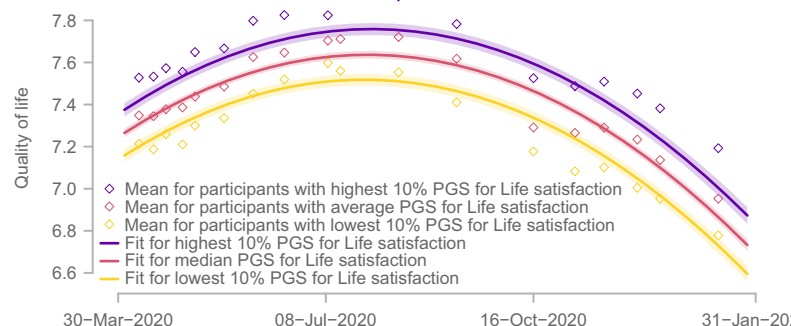

**c) Relative contribution of Life satisfaction PGS increases (meta-analysis p-value: 3.1×10⁻³)**

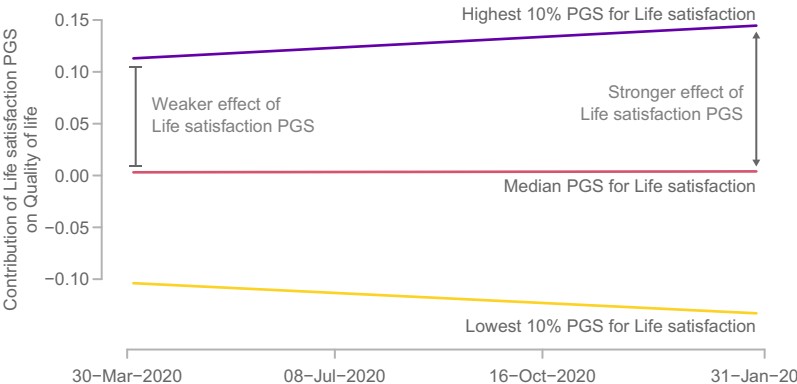

**d) Correlation between Life satisfaction PGS and quality of life is higher in later questionnaires**

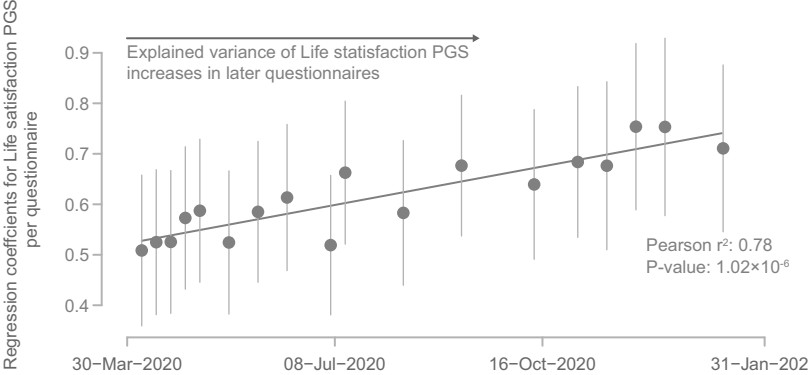

**Fig 3. The genetic contribution to perceived quality of life increased over the course of the pandemic.** a) Course of mean quality of life over time. b) Fitted longitudinal model for quality of life stratified by the PGS for Life satisfaction of the samples genotyped on the Global Screening Array (for other samples see S4 Fig). The shaded area represents the 95% confidence interval for the model fit. c) Contribution of the Life satisfaction-PGS to perceived quality of life over time. The diverging lines indicate the increased importance of the Life satisfaction PGS to perceived quality of life as time progresses. d) Regression coefficients of the Life satisfaction PGS and perceived quality of life over time. The error bars represent the 95% confidence interval for the regression coefficients. This shows an increase in the explained variance of the PGS during the COVID-19 pandemic.

waning impact of this stressor, the effect of genetics became more dominant, reflecting a return to the impact it had prior to the emergence of COVID-19. Unfortunately, we cannot directly test this hypothesis since we have no pre-pandemic measurement on perceived quality of life. However, the *Twins Early Development Study* on adult twins in Great Britain did show that the first month of lockdown did not result in major changes in the genetic or environmental origins in psychological or attitudinal traits compared to a pre-pandemic timepoint [31]. A similar observation was made by the *Netherlands Twin Register*, they did not find a significantly altered heritability for well-being related outcomes before and after the start of the pandemic [32]. Both these studies support the explanation that the increase in the contribution of genetics that we observe is a consequence of the ongoing pandemic rather than a slow return to a pre-pandemic situation.

To the best of our knowledge, we are the first to show an increased relative impact of genetics on wellbeing during the COVID-19 pandemic. In a study in twins from the UK by Rimfeld et al. [33] heritability estimates for depression, general anxiety and other (wellbeing) items were studied, and compared over time at five different timepoints. The first of which was set in 2018, and later timepoints set from July 2020 to March 2021. This study did not show a change in heritability over time, whereas we do observe this effect. This might be explained by the relatively low sample size compared to our study. In addition, the studied samples were strictly confined to young adults in the UK, whereas we have studied a much broader range of Dutch adults. The results of both our studies could therefore coexist together and do not fully contradict each other.

**Host genetic contribution to COVID-19 infections.** A change in the influence of genetics, as seen in PGSs, over time is not restricted to wellbeing outcomes. We also observed a change in the influence of the PGS for COVID-19 susceptibility on actual infections. As expected, we observed a strong increase in the number of infections among participants over the course of our study (Fig 4). At the start of our study in March 2020, only 198 infections within our region had been reported by the government, whereas this number had increased to 7,579 in January 2021 [34]. While the PGS for COVID-19 susceptibility increased the risk of being infected (p-value = $1.28 \times 10^{-22}$), we also observed that the effect of this PGS declined over time (p-value = $5.15 \times 10^{-30}$).

One possible explanation for this is testing bias. At the beginning of the pandemic in the Netherlands, testing was almost exclusively reserved for severely ill patients. It was only in June 2020 that anyone with symptoms could get a test [35] and only in the months following that testing was extended to asymptomatic individuals who had been in contact with someone who had tested positive for a SARS-CoV-2 infection. We suspect that the same testing bias exists in other cohorts of the COVID-19 susceptibility GWAS. Additionally, several cohorts that contributed to the COVID-19 susceptibility GWAS are based on hospitalized patients. Therefore, some of the GWAS signal is likely informative for COVID-19 severity rather than susceptibility, which could explain that the performance of the corresponding PGS diminishes once infected people with few or no symptoms are being widely tested.

Alternatively, it could be that the contribution of host's genetics on COVID-19 susceptibility is different for new SARS-CoV-2 variants. Near the end of our study the more the infectious

**a) COVID−19 infection incidence curve**

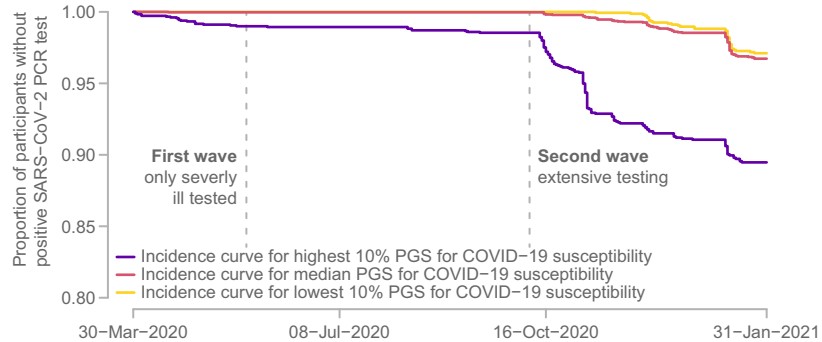

**b) Fitted model in 12,541 global screening array samples**

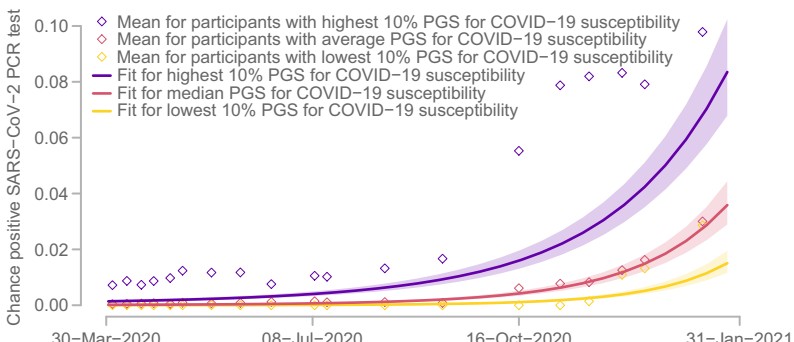

**c) Relative contribution of COVID−19 susceptibility PGS decreases (meta-analysis p-value: 5.15×10$^{-30}$)**

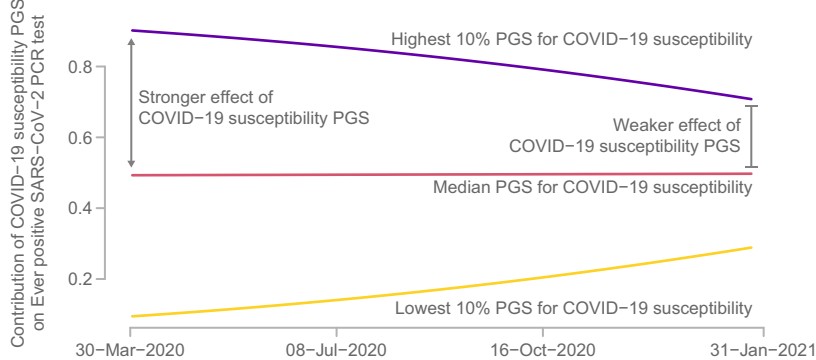

**d) Correlation between COVID−19 susceptibility PGS and infections is lower in later questionnaires**

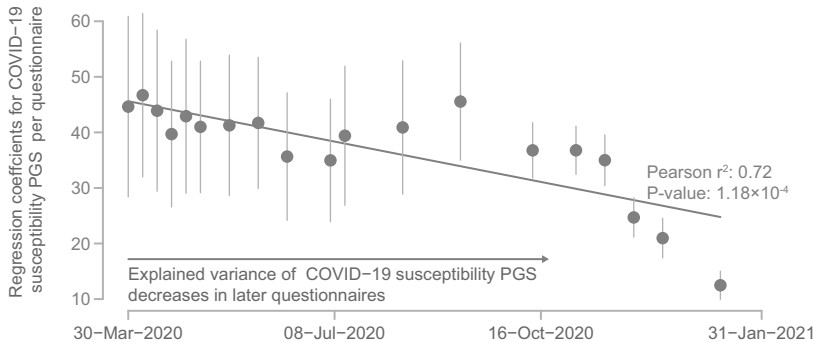

**Fig 4. COVID-19 incidence increases over time with a decreased importance of genetic COVID-19 susceptibility.**
a) Kaplan-Meier curve stratified by genetic risk for COVID-19 infections. b) Fitted logistic longitudinal model for
COVID-19 shows increasing infections regardless of genetic risk. The shaded area represents the 95% confidence
interval for the model fit. c) The contribution of COVID-19 susceptibility to the risk of having had COVID-19 slowly
decreased over time. d) Regression coefficients of the COVID-19 susceptibility PGS and ever testing positive for
SARS-CoV-2 by PCR test over time. The error bars represent the 95% confidence interval for the regression
coefficients. The plot shows a decrease in the variance explained by the PGS during the COVID-19 pandemic.

B.1.1.7 (Alpha) variant was in the process of becoming the dominant variant in the Nether-
lands, this might also explain why the PGS of the COVID-19 susceptibility GWAS was becom-
ing less informative.

As we discuss above, this probably reflects a different mechanism; a shift in who tests posi-
tive over time compared to changes in wellbeing items. This does support to the idea that these
interaction effects are real and need to be considered when applying PGS in other contexts.

## Sensitivity analysis of longitudinal models

We used three different strategies to validate the 11 outcomes vs PGS interactions that we
found using the longitudinal mixed-effects models. First, to rule out that these observed effects
are caused by attrition bias we also performed analyses on a subset of samples, totalling 7,502
samples, who had completed questionnaires at 1, 4, 7 and 10 months (S2 Note and S9 Table).
Using this subset, we could test 10 out of the 11 interaction effects, one model did not converge
in this subset and could not be validated. We found that all interactions show the same effect
direction and that four are Bonferroni significant within this subset. This shows that non-ran-
dom dropouts of participants are not driving our findings.

Secondly, we calculated the association between the PGS-outcome pairs at each timepoint
separately and subsequently tested whether the strength of this association changed over time
(S11 Table and S8 Fig). While such models are simplistic and underpowered, they do allow us
to confirm the validity of the results from our mixed-effects models. Herein we were able to
Bonferroni significantly replicate 9 out of 11 interactions.

Finally, in the mixed-effects models we rely on PGSs as a basis for the variance that is
explained by genetics. However, here it is important to note that a change in the variance
explained by a single PGS may not necessarily reflect all the impact of genetics. Gene-environ-
ment interactions may cause the PGS to be less effective while the total contribution of genetics
as a whole stays constant. Considering this, we aimed to validate our results independently
from PGSs. We have done so by estimating the heritability of each of the seven outcome items
for which we observed a variable genetic contribution over time in our longitudinal models
(S13 Table). We were able to estimate the heritability for each timepoint separately for six out-
come items, and then tested whether the heritability increased or decreased over time. We
were not able to reliably estimate the heritability for all timepoints for the 'Ever positive
SARS-CoV-2' item since we did not have enough cases at the start of the pandemic. For four
outcome items ('Felt good', 'Felt physically exhausted', 'Felt tired', and 'Was easily tired', p-
value $\leq 0.05$) we observed a significant increase of heritability over time (S10 Fig) without sig-
nificant changes to the contribution of shared environment over time (S11 Fig). The two other
outcomes ('Concerned about the COVID-19 pandemic' and 'Quality of Life') could not be val-
idated using this method. This might be the result of limited power as the heritability estimates
are small and have a relatively large standard error. Given that all the effects are in the expected
direction, and that we can convincingly validate the increased genetic contribution on four
outcome items, we are confident that the results obtained using the mixed-effects models and
the PGSs are reliable.

## Limitations

While the Lifelines cohort is representative of the general population [1], not all of the Lifelines participants decided to fill in our COVID-19 questionnaires and we could only include samples that had been genotyped. Both introduced a sample bias compared to the whole cohort. For instance, the average PGSs of some traits were different between genotyped individuals that did not take part in this study compared to the 27,537 samples used in our baseline analysis (S9 Fig), indicating a small genetic bias for the willingness to fill-in the COVID-19 questions. In addition, some of the answers to our baseline outcomes were also different for our sample set compared to the participants that were not genotyped (S12 Table), indicating a bias in the selection of samples that where genotyped. It is therefore possible that our estimated effect sizes do not exactly reflect the effects in the general population.

It should be noted that the samples used in this study are also part of the GWASs for BMI, educational attainment, and COVID-19 susceptibility (S3 Table). For the BMI and educational attainment GWASs this only holds for the samples genotyped using the HumanCytoSNP-12 array. Here, we found that all significant associations are also nominally significant (p-value $\leq 0.05$) and have the same direction when only using the samples not included in the BMI and educational attainment GWASs. Most effects, 83 out of the 89, are even Bonferroni significant using only the GSA data. For the COVID-19 susceptibility we cannot rule out that the baseline association between the PGS and infection status is biased by the overlapping samples. However, this does not explain the change in PGS performance over time.

## Implication

Although the genetic effect on wellbeing became stronger over time, its contribution remains relatively small. The effect sizes of interaction effects are small as well. However, the accuracy of PGSs is currently still limited, which likely has hampered our ability to find larger effects. In reality, the effect of genetics on each of these traits is expected to be larger. Furthermore, our findings do demonstrate that it is not only the presence or absence of environmental stimuli that can modulate the effect of genetic variants, the duration of a stimulus in conjunction with a genetic background can also modulate the outcome. Accounting for this by the use of longitudinal biobanks can help improve the power of GWASs to detect associated variants [36], enable more accurate risk predictions using PGSs and allow more accurate patient stratification [37], and provide insights into the complex interplay of genetic predisposition and environment on diseases and traits [38].

## Conclusion

We have been in the unique position to observe a synchronized and prolonged exposure to a shared continuous stress factor and an increasingly abundant infectious disease. This allowed us to observe longitudinal changes in the relative contribution of nature and nurture on wellbeing and COVID-19 infections. Our results indicate that participant's responses to the COVID-19 pandemic were at least partially driven by their genetic predisposition and that this genetic contribution changes over time.

## Methods

### Ethics statement

The Lifelines study was approved by the ethics committee of the University Medical Center Groningen, document number METc2007/152. All participants signed an informed consent form prior to enrolment.

## Cohort

We selected study participants from the Lifelines COVID-19 cohort [2], for which 139,713 adult participants with a known email address were approached from the Lifelines population cohort [1]. Lifelines is a prospective multigenerational population cohort following 167,000 individuals in the three northern provinces of the Netherlands (Drenthe, Friesland, Groningen) since 2006, and collects detailed information and biological samples from its participants (Fig 1). The repeated questionnaires sent out to the Lifelines COVID-19 cohort include items about sociodemographic parameters, chronic diseases, COVID-19 infection, general health and symptoms, medication use, mental health, well-being, social life and lifestyle. At least once a month, participants were invited to respond to these questionnaires. In total, 19 questionnaires were sent out between March 30, 2020 and January 5, 2021. On average, 39,066 participants responded to each questionnaire (S1 Table).

## Genetic data and PGSs

Within the Lifelines population cohort 36,339 participants have been genotyped using the Global Screening Array. An additional 14,463 participants have been genotyped using the HumanCytoSNP-12 chip, making the total number of genotyped individuals 50,802.

Genotyped individuals were filtered to only include samples of European ancestry according to principal component analysis. Genotype data from arrays were separately imputed on the Sanger imputation server using the Human Reference Consortium reference panel [39].

We calculated PGSs by applying PRS-CS [40] with our defined set of traits (S3 Table). We did this separately for each of the two genotyping arrays. For each of the traits, we downloaded the complete summary statistics from the indicated source. To comply with the PRS-CS input format, we added reference SNP identifiers (RSIDs) to the GWAS summary statistics, when these were not initially present. This was done by matching genomic locations for each of the variants to those from dbSNP [41]. For the genotype data generated with the HumanCytoSNP array, RSIDs were matched to the genotype data when the genomic location and both the alleles matched variants from dbSNP (build 152). This was not necessary for the genotype data generated with the GSA. Thereafter, we excluded SNPs with ambiguous alleles. We also removed SNPs with a minor allele frequency below 0.01, an imputation score below 0.3, or a missing call rate greater than 0.25. Data was converted to bpgen format using PLINK 2.0 to maintain allelic dosage information [42,43].

PRS-CS accounts for inaccuracies in effect sizes and linkage disequilibrium (LD) patterns using a reference panel and through shrinkage of effect-size estimates. It does not require pruning or thresholding of variants. PRS-CS was run on each autosome separately using the processed PLINK datasets and the complete GWAS summary statistics. We used the European reference LD panel provided by the PRS-CS authors and the default for the other parameter options. PLINK 2.0 was used to sum variant dosages from the PLINK datasets, weighted by the posterior effect sizes calculated by PRS-CS. Finally, we summed the PGSs that were calculated separately for each of the autosomes and scaled the outcomes to have a mean of 0 and a standard deviation of 1.

## Questionnaire quality control

Of the 50,802 genotyped Lifelines participants, 27,537 participants completed at least one COVID-19 questionnaire and are of European descent. To ensure that our questionnaires were filled in reliably, we performed a principal component analysis (PCA) for each of the 19 questionnaires to detect spurious signals. To enable this quality control, we first selected questions that were answered by at least 95% of the participants of that questionnaire. This resulted in at least 98 questions per questionnaire that were answered by nearly all participants.

Subsequently, we calculated the per participant missing rate for each questionnaire for these questions. For each participant, we excluded the questionnaires with more than 5% "missingness". The answers to questions with a maximum of 5% missingness were normalized to have a mean of 0 and a standard deviation of 1. At this point, we still had some missing values in our data, which prevented us from performing a simple PCA. There are many ways to resolve this, but for the purpose of quality control, we decided to simply fill missing answers with the mean answer for that question. Subsequently, we normalized each sample to have a mean of 0 and standard deviation of 1. Finally, we performed a PCA on the sample correlation matrix. This did not detect any outliers with a standard deviation larger than 4, suggesting the absence of respondents who filled out the questionnaires with erroneous answers. As such, we did not need to remove samples from the dataset.

Next, to include participants with long-term follow-up, we selected participants who had completed at least one questionnaire in the first half of our study and at least one questionnaire in the second half (before and after August 31, 2020, respectively). Using these criteria, we ultimately included 17,831 participants who had completed an average of 13 out of the 19 questionnaires (S1 Fig and S1 Table).

We assessed the effects of quality control and sample exclusions on the PGSs by comparing the participants who were included in this study to those who were invited to take part in the Lifelines COVID-19 questionnaires but were not included. These either did not respond or did not pass quality control. Welsh's t-tests are used to assess whether differences were significant. For seven traits in the samples genotyped using the Global Screening Array (GSA), and for three traits in the samples genotyped using the HumanCytoSNP-12 array, the PGSs of the included samples in longitudinal analyses are different compared to the samples that were invited (S9 Fig). The figure also demonstrates that sample selection based on quality control has been favourable for controlling false positives caused by loss to follow-up.

## Outcome items

We recoded questionnaire items to a binary, ordinal or continuous variable when this was not already the case (S5 Table). We made sure that the ordering of the ordinal and dichotomous answers corresponded to the directionality implied by the given label or question. Additionally, we removed answer options that did not fit on the ordinal scale, like answers indicating that the question was not applicable to the individual. Questionnaire items with an approximate continuous scale were analysed as continuous variables, whereas items with a more limited number of answer options or for which the answers were not approximately normally distributed were analysed as ordinal or binary variables. After recoding, we removed all the questions where the most frequent answer was given by more than 99% of the participants.

In addition to single questionnaire items, we derived a set of additional outcome items. Current depressive episode was calculated using the decision tree from the standardized "Mini-International Neuropsychiatric Interview (M.I.N.I.)" version 5.0.0. questionnaire manual [44]. We excluded the question "Did you feel tired or without energy almost every day?" because this question was not available in all questionnaires. We also excluded the timepoints for which the question "Did you repeatedly consider hurting yourself, feel suicidal, or wish that you were dead?" was not available.

We determined whether participants were ever tested positive for a SARS-CoV-2 infection with a PCR test. For this, we used a series of questions asking if a PCR test was performed and if the test result was positive or negative. The resulting "ever positive SARS-CoV-2 PCR test" variable was defined per participant and set as *True* for all following questionnaires, starting from the first self-reported positive SARS-CoV-2 PCR test.

BMI was calculated using the most recent self-reported body weight in the questionnaires and the height that had been measured during the most recent physical visit. Weight values below 20 kg and above 220 kg were removed. Values with an absolute difference of more than 20 kg compared to the last previous questionnaire were also removed. The first value was compared to the second and third value and was removed if the absolute difference to the second value was more than 20 kg and that to the third value was more than 30 kg. If only two questionnaires were completed, both points were removed if the absolute difference was more than 20 kg.

## Covariates

We used the participant's sex as registered in the Personal Records Database [45]. Their age, derived using date of birth and date of questionnaire completion, was available for every questionnaire. We used the age at the time of the most recently completed questionnaire. We also assessed if participants live alone or not. This was defined as having zero household members, which we determined using the most recent self-reported number of household members. We also determined whether participants have children living at home using the question on whether participants have children and the most recent number of household members reported below 18 years of age. Chronic illness was extracted from the most recent self-reported chronic disease item, based on the question: 'Do you have a chronic health condition?'. Time was defined as days since March 30, 2020.

## Baseline associations

To identify correlations between responses to questionnaire items and PGSs, we fitted models between all PGSs and both the questionnaire items and derived items within 27,537 participants. First, we filtered the questions for which the answers were not directly influenced by the participant. For example, we removed questions about choices made by their employer. After manual selection, we removed all the questions that were answered by less than 50% of the participants of the questionnaire in which that particular question was asked.

For each question, we selected the appropriate regression model. We chose a normal linear regression model for normally distributed outcomes, a logistic model for binomial data and an ordinal logistic regression model for ordered categorical data.

The models were fitted using the 'statsmodels' package. The quantitative questions were fitted using the OLS class, the binominal questions were fitted using the Logit class and the ordinal questions were fitted by the 'OrderedModel' class. For the 'OrderedModel', we set the 'distr' option to logit, set the maximal number of iterations to 10,000 and set the optimizer to the Broyden-Fletcher-Goldfarb-Shanno (BFGS) method. The models were fitted for each PGS and each question individually while adjusting for sex, age, $age^2$, chronic illness, living alone and children living at home.

The models were fitted separately for each genotype array and the regression coefficients and p-values were combined with the inverse-variance weighting method. Afterwards, the p-values were translated to Z-scores and filtered on significance using a Bonferroni-corrected α of 0.05.

## Longitudinal models

For all PGS–question pairs for which we identified a significant baseline association, we tested if this effect was stable over time within 17,831 participants. There are 237 PGS-question pairs for questions that did have longitudinal data. For 46 PGS-question pairs we were able to fit mixed-effects models with a random intercept, time and $time^2$ per participant (except for the

model for "ever positive SARS-CoV-2 PCR test" in which we only used a fixed intercept). To investigate differences in changes over time, we included interaction with time for all included variables:

$$y = \beta_0 + \beta_{1p} \times PGS + \beta_2 \times PGS \times time + \beta_3 \times time + \beta_4 \times time^2 + \beta_{5c} \times confounders + \beta_{6c}$$
$$\times confounders \times time + u_s + \varepsilon$$

Herein, $u_s$ and $\varepsilon$ denote the sample random effect intercept and the residual error. Time was denoted as days since March 30, 2020. We included time and time$^2$ to model the non-linear changes over time and adjusted for sex, age (mean-centred), age$^2$, chronic illness (0/1), living alone (0/1) and children living at home (0/1).

We used the lme function from the nlme package for questions with a quantitative answer. Here, we used the optim optimizer [46]. For "ever positive SARS-CoV-2 PCR test", we used the R GLM function with binominal logit link function [47]. The other questions, which have binary outcomes, were fitted using the glmer function with the binominal logit link function and nAGQ = 0 parameter from the lme4 package [48].

Since our cohort was genotyped on two different genotyping arrays, we fitted the model above for each array separately. We used inverse-variance weighting to combine the estimates and standard errors for each term in the model [49]. These where then used to calculate the combined Z-score and p-value that we used to determine the significance of each term.

## Sensitivity analyses

To assess that our findings obtained using the longitudinal mixed-effects models were not driven by non-random dropout of participants, misspecification of our model, or the nature of the PGSs, we have done three sensitivity analyses.

**Attrition bias sample subset.** We selected 7502 samples that all completed questionnaires 4, 9, 14, and 19 to create a sample that is guaranteed not to suffer from attrition bias or non-random censoring. Using only this subset of samples with answers given only at these four time points we reran all longitudinal models for which we found a significant PGS × time interaction.

**Validity of mixed-effects models.** We performed a sensitivity analysis using separate linear and logistic models for each questionnaire item on each individual timepoint for the 17,831 samples with longitudinal data. The resulting regression coefficients of the PGSs for each timepoint were subsequently correlated with time to analyse if the effect of the PGSs on the different (derived) outcomes changed over time. For each timepoint and each item, we used linear, logistic, or ordinal regression models which were also used by calculating the baseline associations, including recoding criteria and parameters for the models and with adjustment for sex, age, age$^2$, chronic illness, living alone and children living at home. The regression coefficients of the PGSs were extracted from the models and grouped per question. These models were fitted separately for each genotype array and the regression coefficients were combined with the inverse-variance weighting method. Subsequently, for each PGS and each question, we calculated the Pearson correlation between PGS regression coefficients and time to analyse if the effect of the PGS on the answer to the question changed over time. For each question, time was defined as the number of days that had passed when the questionnaire containing the relevant question was sent out, since March 30, 2020.

**Heritability estimates.** We also wanted to validate the altered impact of genetics independent of the PGSs. Therefore, we calculated heritability estimates and the variance explained by shared environment based on pedigree information and household composition. This was done for each of the available timepoints, the total and phenotypic variances were estimated

from the linear mixed model using the ASReml-R package. This analysis was also performed in the 17,831 samples with longitudinal data. Narrow-sense heritability was calculated using pedigree information as:

$$h^2 = \sigma_a^2/(\sigma_a^2 + \sigma_f^2 + \sigma_e^2)$$

and the proportions of variance explained by shared family environment was calculated as:

$$c^2 = \sigma_f^2/(\sigma_a^2 + \sigma_f^2 + \sigma_e^2)$$

Herein, $\sigma_a^2$ Is the additive genetic variance, $\sigma_f^2$ is the shared environmental variance, and $\sigma_e^2$ is the residual variance. All estimates were adjusted for sex, age, age$^2$, chronic illness, living alone and children living at home. To test the significance of the $h^2$ estimates ($h^2 > 0$), the model in which all variances were estimated was compared to a model in which additive genetic variances was constrained to be zero using a likelihood-ratio test. Shared environment was determined by assessing whether individuals lived in the same household at the baseline assessment in Lifelines.

Within these estimates we subsequently attempted to model longitudinal effects. For each of the outcomes, we modelled the heritability using time as the independent variable in a linear model. Herein, time was defined as the number of days that had passed when the questionnaire containing the relevant outcome was sent out, since March 30, 2020. The same models were applied to variances explained by shared environment.

## Correlation with nationwide statistics

We collected a number of publicly available datasets that we hypothesized might be able to explain part of the variation in the perceived quality of life over the course of the pandemic. We collected three variables that convey the state of the pandemic in the Netherlands: the confirmed COVID-19 cases per day published by the Dutch National Institute for Public Health and the Environment (RIVM) [34], the intensive care unit (ICU) occupancy by COVID-19 patients published by the 'Landelijk Coördinatiecentrum Patiënten Spreiding' (LCPS) [50] and, as a measure of the severity of lockdown in the Netherlands, we downloaded the 'Stringency Index' from the Oxford COVID-19 Government Response Tracker [51]. Additionally, we retrieved the relative change in work- and recreational-related mobility from the Google COVID-19 Community Mobility Reports [52]. Finally, we downloaded a dataset on the weather per day from The Royal Netherlands Meteorological Institute for a central Northern Netherlands weather station (Eelde, Drenthe) [53].

A 7-day moving average was calculated for the confirmed COVID-19 cases, the relative change in both work and recreational mobility, the hours of sunshine per day and the average temperature over 24 hours (in degrees Celsius). These variables are visualized together with the ICU occupancy by COVID-19 patients and the Stringency Index (S6 Fig). Together with the publicly available variables, this figure presents the mean perceived quality of life per questionnaire over the average response date for that questionnaire.

For each of the visualized variables, we extracted the values that coincided with the average response dates for the questionnaires. Thereafter, the Pearson correlation was calculated between each of these variables and the mean perceived quality of life (S7 Fig and S10 Table).

## Supporting information

**S1 Table. Used questionnaires.** 19 questionnaires were sent from March 2020 to January 2021. The interval varies from weekly to monthly. The number of samples that we used in

baseline and in longitudinal analyses are represented in columns 5 and 6 respectively.
(XLSX)

**S2 Table. Sample descriptions.** Descriptive statistics for the two sample selections in our cohort. 27,537 baseline samples were used for baseline associations, and 17,831 samples were used in the longitudinal analysis. We show the total number of samples, the distribution of age and BMI, and the frequencies of values for the other covariates. Chronic illness was extracted from the most recent self-reported chronic disease item.
(XLSX)

**S3 Table. Used polygenic scores.** The PGSs for the traits and diseases that were assessed.
(XLSX)

**S4 Table. Genetic correlations of selected traits.** The output produced by Linkage disequilibrium score regression for each combination of selected traits.
(XLSX)

**S5 Table. Baseline question overview.** The overview of questions that have been tested for associations with polygenic scores. The full questions and answers have been translated into English from Dutch. Some questions are specific to the context of a broader question or heading. In these cases, the overarching question or heading is separated from the specific question by a '/'. The third column represents how the individual answers have been coded in the regression model.
(XLSX)

**S6 Table. Baseline association Z-scores.** The Z-Scores for baseline associations between polygenic scores (columns), and outcome items (rows). Some questions are specific to the context of a broader question or heading. In these cases, the overarching question or heading is separated from the specific question by a '/'.
(XLSX)

**S7 Table. Baseline association p-values.** The p-values for baseline associations between polygenic scores (columns), and outcome items (rows). Some questions are specific to the context of a broader question or heading. In these cases, the overarching question or heading is separated from the specific question by a '/'.
(XLSX)

**S8 Table. Longitudinal question overview.** The overview of questions that have been tested for interaction effects between polygenic scores and time. Some questions are specific to the context of a broader question or heading. In these cases, the overarching question or heading is separated from the specific question by a '/'.
(XLSX)

**S9 Table. Results of longitudinal models.** Significance of PRS × Time interaction term. Models that did not converge to a solution are omitted.
(XLSX)

**S10 Table. Correlation between mean perceived quality of life with 7 publicly available variables.** The Pearson correlations calculated between the mean perceived quality of life and 7 publicly available variables. From these variables values were sampled at the average response dates of the questionnaires.
(XLSX)

**S11 Table. Results of the sensitivity analysis.** Results of the sensitivity analysis for the over-time interaction between a question from the questionnaire and the PGS trait.
(XLSX)

**S12 Table. Comparison of samples that have been genotyped and samples that have not been genotyped.** A comparison of samples that have been genotyped and samples that have not been genotyped based on all significant baseline questions. Questions that in our baseline analysis have been considered continuous were tested using a t-test. All other questions were analysed using a Fisher's exact test. For these questions simulated p-values (based on 1e+07 replicates) were used when there were more than two answer options. We applied a Bonferroni correction for all questions based on 143 questions.
(XLSX)

**S13 Table. Heritability estimates and estimates of explained variance due to shared environment.** The estimated narrow-sense heritability (expressed as h2), and variance explained by shared environment. Shared environment was determined by assessing whether individuals lived in the same household.
(XLSX)

**S1 Fig. Number of completed questionnaires per participant used for the longitudinal models.** The 17,831 selected participants completed on average 13 questionnaires ranging from 2 to 19.
(PNG)

**S2 Fig. PGS correlations between traits.** Pearson r correlations between the calculated PGSs of all 27.537 participants.
(PNG)

**S3 Fig. Effect of PGS on outcome items at baseline.** The baseline associations between PGS and the different outcomes. The baseline association are obtained by a meta-analysis over samples run on Global Screening Array and the HumanCytoSNP-12.
(PDF)

**S4 Fig. Visualizations of significant time and PGS interactions.** Each significant interaction between time (denoted in days starting from March 30, 2020) and PGS visualized over time, stratified by the PGS for which the interaction was observed to be significant. The 10th percentile, the median and the 90th percentile illustrate how the PGS interacts with time. On top the complete model for its respective outcome measure is presented. Herein the contributions of all terms are considered, and the temporal aspect of the outcome variable can be seen and compared with the interaction effect. The shaded areas represent the 95% confidence interval of the model fit. On the bottom only the relative contribution of the PGS is taken into consideration. Converging percentile lines indicate that the PGSs have a decreasing effect on the outcome measure. Contrariwise, diverging percentile lines indicate that a PGS has an increasing effect on the outcome measure showing that genetics play an increasingly important role.
(PDF)

**S5 Fig. Correlation heatmap for a set of wellbeing and fatigue items.** The spearman correlation estimates calculated for the wellbeing and fatigue items for which the impact of genetics has significantly increased during the pandemic. The heatmap shows that the items 'Was easily tired', 'Felt tired', and 'Felt physically exhausted' are highly correlated (Spearman's rho = 0.69–0.85). The item 'Felt good is also correlated to these items (Spearman's rho for 'Was easily tired' = -0.60). Quality of life is correlated to the other items as well (Spearman's rho values for

'Was easily tired' and 'Felt fine' are equal to -0.25 and 0.25 respectively). The baseline instance of every question was used. From all 27,537 participants the pairwise complete observations were used to handle missing values.
(PNG)

**S6 Fig. Development of quality of life during the COVID-19 pandemic in the Netherlands.** The rise and fall of the mean perceived quality of life coincides with the development of the COVID-19 pandemic in the Netherlands. The panels 1–3 present the confirmed COVID-19 cases, COVID-19 ICU occupancy, and stringency of the measures respectively. These figures indicate three instances with a large number of COVID-19 patients and a consequent high lockdown stringency. These instances coincide with low and decreasing mean perceived quality of life as can be seen in panel 8. The lockdown stringency is also in part reflected in the retail and recreational-related mobility, and to a lesser degree the work-related mobility (panels 4–5). Panels 6–7 present the increase of the average temperature and the hours of sunshine respectively during the spring and summer months, and their subsequent decrease during autumn and winter. Of these weather-related variables, we observe that the correlation of the mean perceived quality of life and the average temperature is especially strongly correlated (Pearson r = 0.88, p-value = $1.07 \times 10^{-3}$). It is therefore likely that the development in the mean perceived quality of life is in part a seasonal, weather influenced, effect. However, given that mean perceived quality of life is also strongly negatively correlated with the Stringency Index (Pearson r = -0.71, p-value = $1.28 \times 10^{-6}$), we think it is conceivable that the state of the pandemic is equally influencing the quality of life participants are experiencing.
(EPS)

**S7 Fig. Heatmap of quality of life.** The Pearson r correlation estimates calculated within 7 publicly available variables (averaged over 7 days if indicated) and the mean perceived quality of life. For each of the variables, we extracted the values that coincided with the average response dates for the questionnaires.
(PNG)

**S8 Fig. Results of the Sensitivity analyses.** We validated our findings by fitting models for each PGS and each question separately for each questionnaire. The regression coefficients of the PGS were extracted from the models and plotted according to the date of each questionnaire. The error bars represent the 95% confidence interval for the regression coefficients. The plots showed a change in regression coefficient over time which indicates that the PGS explains less or more variance of the question outcome over time.
(PDF)

**S9 Fig. Comparison of polygenic scores between invited and included individuals.** Polygenic scores (PGSs) for the participants that were included in this study compared to those that were invited to take part in the Lifelines COVID-19 questionnaires, but that were not included because they either did not respond or did not pass quality control. The p-values for Welch's t-tests are shown to indicate whether the two groups differ significantly or not. Panel A, C and E show the participants that were genotyped using the Global Screening Array. Panel B, D and F show the participants that were genotyped using the HumanCytoSNP-12 array. Panel A and B show the 27,537 baseline samples compared to all other invited samples. Panel C and D show the 17,831 samples used in longitudinal analysis compared to all other invited samples. From these, 10 out of 34 p-values are smaller than an a priori Bonferroni corrected alpha of 0.05. This indicates that a small genetic bias is introduced for the willingness to fill-in the COVID-19 questions. Panel E and F show the 27,537 baseline samples compared to the 17,831 samples used in longitudinal analysis. Herein, the differences in Educational attainment

and schizophrenia are highly significant. Should all baseline samples be included in the longitudinal analysis, such differential attrition would have biased our results. This suggests that selecting a confined set of samples for longitudinal analysis was appropriate.
(PDF)

**S10 Fig. Heritability estimates over time.** Narrow sense heritability estimates ($h^2$) modelled over time show a significant increase for 4 out of 6 outcome items. In each of the panels the annotated $r^2$ depicts the explained variance of the model, while the p-value represents the p-value of the effect size of the time variable. Error bars represent the standard errors of the heritability estimates. The blue line and shaded area represent the fit and the standard error of the linear model respectively.
(EPS)

**S11 Fig. Explained variance of shared environment over time.** Explained variances of environment ($c^2$) based on shared households modelled over time for 6 outcomes. No significant effect of time is observed. In each of the panels the annotated $r^2$ depicts the explained variance of the linear model, while the p-value represents the p-value of the effect size of the time variable. Error bars represent the standard errors of the estimated values. The blue line and shaded area represent the fit and the standard error of the linear model respectively.
(EPS)

**S1 Note. Effect of the Schizophrenia-PGS on concern about the pandemic.**
(DOCX)

**S2 Note. Results of sensitivity analyses.**
(DOCX)

**S1 File. Authorship list of the Lifelines Corona Research Initiative.**
(PDF)

**S2 File. Authorship list of the Lifelines Cohort Study.**
(PDF)

## Acknowledgments

We thank the UMCG Genomics Coordination Center, the UG Center for Information Technology and their sponsors BBMRI-NL & TarGet for storage and compute infrastructure. We thank Katherine McIntyre for the English editing of our manuscript. The authors wish to acknowledge the services of the Lifelines Cohort Study, the contributing research centers delivering data to Lifelines, all the study participants, and the contributions of the investigators to this study: Raul Aguirre-Gamboa, Patrick Deelen, Lude Franke, Jan A Kuivenhoven, Esteban A Lopera Maya, Ilja M Nolte, Serena Sanna, Harold Snieder, Morris A Swertz, Judith M Vonk and Cisca Wijmenga. The authors wish to acknowledge the efforts of the Lifelines Corona Research Initiative and the following initiative participants: H. Marike Boezen, Jochen O. Mierau, Lude H. Franke, Jackie Dekens, Patrick Deelen, Pauline Lanting, Judith M. Vonk, Ilja Nolte, Anil P.S. Ori, Annique Claringbould, Floranne Boulogne, Marjolein X.L. Dijkema, Henry H. Wiersma, C.A. Robert Warmerdam, Soesma A. Jankipersadsing and Irene V. van Blokland.

## Author Contributions

**Conceptualization:** Patrick Deelen, Lude H. Franke.

**Data curation:** C. A. Robert Warmerdam, Henry H. Wiersma, Patrick Deelen.

**Formal analysis:** C. A. Robert Warmerdam, Henry H. Wiersma, Alireza Ani, Judith M. Vonk, Patrick Deelen.

**Funding acquisition:** H. Marike Boezen, Lude H. Franke.

**Investigation:** C. A. Robert Warmerdam, Henry H. Wiersma, Pauline Lanting, Marjolein X. L. Dijkema, Judith M. Vonk, Patrick Deelen, Lude H. Franke.

**Methodology:** C. A. Robert Warmerdam, Henry H. Wiersma, Pauline Lanting, Marjolein X. L. Dijkema, Judith M. Vonk, Patrick Deelen, Lude H. Franke.

**Project administration:** Patrick Deelen, Lude H. Franke.

**Resources:** H. Marike Boezen, Patrick Deelen, Lude H. Franke.

**Software:** C. A. Robert Warmerdam, Henry H. Wiersma, Patrick Deelen.

**Supervision:** Harold Snieder, Patrick Deelen, Lude H. Franke.

**Validation:** C. A. Robert Warmerdam, Henry H. Wiersma, Judith M. Vonk, Patrick Deelen.

**Visualization:** C. A. Robert Warmerdam, Henry H. Wiersma, Patrick Deelen, Lude H. Franke.

**Writing – original draft:** C. A. Robert Warmerdam, Henry H. Wiersma, Pauline Lanting, Judith M. Vonk, Patrick Deelen, Lude H. Franke.

**Writing – review & editing:** C. A. Robert Warmerdam, Henry H. Wiersma, Pauline Lanting, Alireza Ani, Harold Snieder, Judith M. Vonk, H. Marike Boezen, Patrick Deelen, Lude H. Franke.

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
