## [Decision Letter · Decision Letter 0]

22 Sep 2021

Dear Dr Warmerdam,

Thank you very much for submitting your Research Article entitled 'Increased genetic contribution to wellbeing during the COVID-19 pandemic' to PLOS Genetics.

The manuscript was fully evaluated at the editorial level and by independent peer reviewers. The reviewers appreciated the attention to an important problem, but raised some substantial concerns about the current manuscript. Based on the reviews, we will not be able to accept this version of the manuscript, but we would be willing to review a much-revised version. We cannot, of course, promise publication at that time.

If you decide to revise the manuscript for further consideration at PLOS Genetics, please aim to resubmit within the next 60 days, unless it will take extra time to address the concerns of the reviewers, in which case we would appreciate an expected resubmission date by email to plosgenetics@plos.org.

[LINK]

We are sorry that we cannot be more positive about your manuscript at this stage. Please do not hesitate to contact us if you have any concerns or questions.

Yours sincerely,

Giorgio Sirugo

Associate Editor

PLOS Genetics

Scott Williams

Section Editor: Natural Variation

PLOS Genetics

Reviewer's Responses to Questions

**Comments to the Authors:**

Reviewer #1: The paper reports on PGS prediction for a number of traits that were assessed repeatedly during the COVID-19 pandemic. The sample derives from the Northern part of the Netherlands, which represents a part of the country that was not hit very hard by the virus, although lockdown was implemented in a similar manner as the rest of the country, which might have had differential effects: are there possibilities to collaborate with other biobanks in the Netherlands to test for replication?

The Lifelines study has 167,000 participants, this project included 27,537 participants: are these people with genotype data who returned at least one survey? (how many people were approached? Did the people with and without genotype data differ in the traits that were studied?).

Does the group of people that returned surveys include family members? Did their responses resemble each other? Did familial resemblance change during the pandemic?

There were 17831 Ss with genotype data and at least 2 surveys? (one survey in the beginning and one in the second half of the study). Do the PGS of the Ss without survey information differ from those who completed the surveys? I do not understand how figure S9 demonstrates “that sample selection based on quality control has been favourable for controlling false positives caused by loss to follow-up”.

Genotype data were used to obtain PGS for 17 traits: did the GWA discovery studies include Lifelines participants? (or their family members?). If yes, were LOO results obtained before PGS were constructed? Please note: figure 2 does not report on 17, but on 14 PGS. The choice for the discovery GWAS is not fully explained: why the focus on lifestyle and psychological traits?

Baseline / univariate analyses: on how many Ss, from which survey? (the first survey, any survey?).

The 2 arrays (N is given for the 27537 Ss by array but not for the 17831 Ss?) were imputed separately? Were SNPs for PGS harmonized across arrays? (e.g. same numbers of variants?).

Please explain: “We recoded questionnaire items to either an ordinal or dichotomous data type when this was not already the case”: continuous traits were not analyzed as continuous? (line 325 suggests they were?).

Regression (Line 341): did confounders correct for familial relatedness? Or only sex and age?

What is u(s) in regression equation?

How were missing data handled? (e.g. there were Ss with 11 vs 16 returned surveys? How many surveys were required to be included in the longitudinal analyses, or were all data analyzed?).

Reviewer #2: Increased genetic contribution to wellbeing during the COVID-19 pandemic

PGENETICS-D-21-01054

Warmerdam et al

Comments to the authors:

Immediate thoughts re. this paper are that this is a potentially interesting areas as it challenges the dynamic relationship between environmental and heritable contributions to trait variability using a universal (though variable) exposure. in COVID-19. From a basic point of view, one might anticipate a constriction of heritable variance for a complex outcome like well-being in the presence of an environmental challenge. However, in the presence of genetic contributions to COVID-19 experience itself (infection/outcome/the demographics of lockdown and exposure), there may be theoretical reasons as to why an inflation in heritable contributions to wellbeing. Whilst the authors present an examination of a series of PRS and their performance during the pandemic and longitudinal data available, there is no systematic assessment of the heritability of the outcomes in question here through time. This could be achieved using multiple methods and would give an important umbrella assessment of heritable contributions through time (and potentially by inceptive events during the pandemic).

The work presented looks to examine genetic contributions to wellbeing in a collection of ~25000 (17k with at least one questionnaire) participants with data collected through the course of the pandemic (lifelines). In brief, PRS were derived for a series of complex traits (including BMI, COVID risk and severity, EA, and psycho-behavioural traits) and these were associated with 288 questionnaire items running across the sampling period. This includes interesting association patterns such as that between PRS neuroticism/schiz’/depression and questionnaire outcomes re. personality traits and exhaustion. Further, the authors observe associations between other complex factors (EA, alcohol consumption, risky behaviour) and a series of variables recorded through the pandemic by questionnaire. This is interesting - though difficult to interpret clearly, however it does demonstrate the existence of analytical power across the data set available.

Followup analyses used the longitudinal nature of data collection to examine the possibility of varying genetic contributions to recorded outcomes through time (in those cases where there is association in baseline data). In this analysis, 11 associations showed some time dependence with 2 beating chance. Mean perceived quality of life varied and there was an apparent increase in the association between PRS for life satisfaction with some (who had high PRSs) showing resilience. This effect was reassessed in a sub analysis of 7500 participants with compete data at 4 time points.

The authors also show interesting relationships between COVID-19 PRS (susceptibility) and both incidence of infection (with increased incidence by score) and susceptibility - but with contributions to overall susceptibility (magnitude of effect) going down with time.

These are extremely interesting observations and the authors are conservative with both their interpretation of effect sizes and also the inferences drawn about these variable observations - this is great. However, there are a series of specific queries which need to be addressed:

- Given the nature of the analyses and the PRS being deployed here, the structure of the lifelines sampling frame will have an important effect on the presence and interpretation of estimates. One assumes that the ~25k respondents are not completely representative of the overall lifelines study and it would be good to have more analytical description of the actual sample and discussion on how this might influence outcomes. For example, the analysis of volunteers who are self-selected (a status likely correlated with many of the outcomes being assessed) may have an influence on the association seen through time in this data set.

- How are PRS chosen and constructed?

- Related to the question above, what is the underlying correlation structure for heritable contributions to the risk factors (PRS) and outcomes being assessed? This is difficult for the questionnaire outcomes, but for the PRS, it is possible to generate estimates of shared heritability across the traits taken into PRS exposures and to ask about the relative independence of these. Ie. what is the rg across the GWAS that formed the PRS?

- The analysis of COVID-19 risk and or severity genetics is tricky. As to the former, whilst case control analysis may be possible, the nature of the cases (in particular) will be important here re. the changing nature of COVID-19 case status through time in this data set. It is likely the case that the cases in the later stages of he longitudinal data here are substantively different to those at the start (when the HGI was run) and that these differences lend to a lessening of overlap between the heritable contributions to case at later stages of the pandemic (notwithstanding the complications of vaccination). Further, the analysis of severity is necessarily a case-only assessment. Again, features which are associated with case status are likely also to be associated with the other questionnaire derived variables here. As a consequence, when stratifying GWAS on case status to undertake severity analyses in the HGI, biases can be introduced which alter the interpretation of the PRS severity and therefore the downstream associations with lifelines variables. These types of complexity do need to be considered in this paper as they will not only influence baseline associations, but also are likely to influence the longitudinal associations see in this work.

- One technical point - whilst the trends in PRS association seen are compelling, they are not so strong as to overcome the limited power in these analyses. Correlations with time are fascinating here, but there should be emphasis put on the scale of these differences (which is there I concede) and also the power of the analyses. Replication would of course be excellent - especially as there are a series of large studies with comparable - cross-pandemic - longitudinal data and genetics.

- Lastly - referring to my point above - if it is possible to assess variation in heritable contributions to questionnaire responses and or exposures in this sample through time, then a coincident assessment of the variation in PRS effects but he same time points (i.e. superimposed onto heritability changes) would be an excellent addition.

Taken together, this is an interesting field and one which could take advantage of unique conditions.

Reviewer #3: The main objective of the manuscript is to investigate the effect of genetic predisposition to certain traits and conditions on the physical and mental health of individuals affected by the COVID-19 pandemic. They conducted a study on 17,831 participants from Lifeline’s biobank with genetic data (total N=167,000). Nineteen repeated questionnaires were sent out to the participants between March 30, 2020 and January 5 2021. This included total of 288 questions related to sociodemographic parameters, chronic diseases, COVID-19 infection, general health and symptoms, medication use, mental health, well-being, social life, and lifestyle. They computed polygenic scores on 17 traits and conditions by applying PRS-CS methods on GWAS summary statistics of these traits. Then, they conducted series of analyses to test for association between 1) PGSs and baseline questionnaire responses. 2) Longitudinal models on the significant PGS-Question from baseline. The analysis with the baseline questionnaire identified 30 significant associations with the most significant association between Neuroticism-PGS and “depression, schizophrenia, felt nervous, felt tired quickly”, Life satisfaction-PGS, and “excessive worrying, quality of life”. Other interesting finding includes a positive association between educational attainment and trust in the government response to COVID-19. Longitudinal models identified 11 FDR significant associations, and authors mainly focused on Life satisfaction PGS and perceived quality of life in the paper. It is an exciting finding showing that individuals with higher PGS for Life satisfaction have increased perceived quality of life compared to individuals with low PGS for life satisfaction even though mean the perceived quality of life decreased at the end of their follow-up period.

Main comments:

• Authors have presented an interesting study that gives insight on the impact of the COVID-19 pandemic on the perceived quality of life. Overall results and data are well presented, and the manuscript is written well.

• Although Figure 2 shows the association results, it would be great if Zscore could be made available in the supplementary files.

• Models are adjusted for chronic conditions. However, there was no mention of what and how the chronic disease was selected? Is BMI included in the chronic condition? As obesity has a high correlation with depression. I would assume adjusting the model with BMI would be necessary.

• Since the genotype data was generated on two different platforms, the authors said they performed the analysis on the two sets separately and then performed a meta-analysis. If the genetic data was imputed already, why didn’t author consider merging the data and run as a single set rather than a meta-analysis? This will increase the power. Of course, they need to adjust for the genotyping panel – which is standard practice in the field.

• Figure 3b and 4b shows the PGS vs Quality of life for 12,522 samples. However, earlier in the paper, they state ~17 samples were selected for the study. Can they clarify the discrepancy? Is the rest of the data in these two figures are from 12k samples or 17k?

• The paper does not follow the PLoS gen format. There is no introduction section in the paper. Authors should revise the paper and adhere to formatting guidelines.

• The discussion section is surprisingly short. My suggestion would be to restructure the paper and move several discussion points from the results section to the discussion.

• Also, highlighting limitations and future directions for the paper would be great. There are several typos across the manuscript and figure. For example, Typo in figure 4b

**Have all data underlying the figures and results presented in the manuscript been provided?**

Reviewer #1: **No: **Not entirely clear what the procedures are to access the data (through Life Lines in the Netherlands?)

Reviewer #2: Yes

Reviewer #3: Yes

PLOS authors have the option to publish the peer review history of their article (what does this mean?). If published, this will include your full peer review and any attached files.

Reviewer #1: No

Reviewer #2: No

Reviewer #3: **Yes: **Anurag Verma

---

## [Decision Letter · Decision Letter 1]

15 Feb 2022

Dear Dr Warmerdam,

Thank you very much for submitting your Research Article entitled 'Increased genetic contribution to wellbeing during the COVID-19 pandemic' to PLOS Genetics.

The manuscript was fully evaluated at the editorial level and by independent peer reviewers. The reviewers appreciated the attention to an important topic but identified some concerns that we ask you address in a revised manuscript. Specifically, prior to publication we request that you amend your manuscript and briefly address the concerns raised by referee # 1, in particular the one regarding the non inclusion of Lifelines samples/data in COVID-19 susceptibility GWAS.  Please explicitly state how that may or may not affect results in your discussion.

We therefore ask you to modify the manuscript according to the review recommendations. Your revisions should address the specific points made by each reviewer.

[LINK]

Yours sincerely,

Scott M. Williams

Section Editor: Human Variation

PLOS Genetics

Scott Williams

Section Editor: Human Variation

PLOS Genetics

Reviewer's Responses to Questions

**Comments to the Authors:**

Reviewer #1: The authors have responded well to some of the questions and concerns that were raised. However, their reply to the construction of PRS raises concerns about the validity of their approach and hence the results in the paper.

The question was: "Genotype data were used to obtain PGS for 17 traits: did the GWA discovery studies include Lifelines participants? (or their family members?). If yes, were LOO results obtained before PGS were constructed?"

Reply: "Lifelines was one of the 126 studies included in the BMI GWAS, and contributed approximately 1% of the total samples for the COVID-19 susceptibility GWAS. Since the Lifelines samples represented, for each of these GWASs, only a very small part of the total number of samples, we expect that the effect of this on our PGSs is very limited. We therefore did not use LOO results before constructing the PGSs."

The expectation of the authors is unfounded. Please consult: Tutorial: a guide to performing polygenic risk score analyses. Choi SW, Mak TS, O'Reilly PF. Nat Protoc. 2020 Sep;15(9):2759-2772. doi: 10.1038/s41596-020-0353-1 and the references therein.

I cannot asses from the authors' reply: were BMI and covid-19 susceptibility the only two discovery GWAMA's in which Lifelines participated? (this seems unlikely).

Reviewer #3: The revised version of the manuscript is much improved and authors have addressed all the comments aptly.

Minor comment: Figure S11 is not mentioned in the main text.

**Have all data underlying the figures and results presented in the manuscript been provided?**

Reviewer #1: Yes

Reviewer #3: None

PLOS authors have the option to publish the peer review history of their article (what does this mean?). If published, this will include your full peer review and any attached files.

Reviewer #1: No

Reviewer #3: No

---

## [Editor Report · Decision Letter 2]

7 Mar 2022

Dear Dr Warmerdam,

We are pleased to inform you that your manuscript entitled "Increased genetic contribution to wellbeing during the COVID-19 pandemic" has been editorially accepted for publication in PLOS Genetics. Congratulations!

Yours sincerely,

Giorgio Sirugo

Associate Editor

PLOS Genetics

Scott Williams

Section Editor: Human Variation

PLOS Genetics

Comments from the reviewers (if applicable):

**Data Deposition**

http://datadryad.org/submit?journalID=pgenetics&manu=PGENETICS-D-21-01054R2

**Press Queries**

---

## [Editor Report · Acceptance letter]

21 Apr 2022

PGENETICS-D-21-01054R2 

Increased genetic contribution to wellbeing during the COVID-19 pandemic 

Dear Dr Warmerdam, 

We are pleased to inform you that your manuscript entitled "Increased genetic contribution to wellbeing during the COVID-19 pandemic" has been formally accepted for publication in PLOS Genetics! Your manuscript is now with our production department and you will be notified of the publication date in due course.

With kind regards,

Livia Horvath

PLOS Genetics

On behalf of:
